# Physiological responses and adaptations to exercise training in people with or without chronic obstructive pulmonary disease: protocol for a systematic review and meta-analysis

Johan Jakobsson  , Jana De Brandt  , André Nyberg 

Department of Community Medicine and Rehabilitation, Physiotherapy, Umea University, Umea, Sweden

**Correspondence to**
Mr Johan Jakobsson;
johan.jakobsson@umu.se

## ABSTRACT

**Introduction** Exercise training is a cornerstone in managing chronic obstructive pulmonary disease (COPD), leading to several physiological adaptations including, but not limited to, structural and muscular alterations, increased exercise capacity and improved cognitive function. Still, it is not uncommon that the acute physiological response to an exercise session and the extent of chronic adaptations to exercise training are altered compared with people without COPD. To date, potential differences in acute physiological responses and chronic adaptations in people with or without COPD are not fully understood, and results from individual studies are contradictory. Therefore, the current study aims to synthesise and compare the acute physiological responses and chronic adaptations to exercise training in people with COPD compared with people without COPD.

**Methods and analyses** A systematic review of randomised controlled trials (RCTs), non-randomised studies of interventions (NRSIs) and cross-sectional studies (CSSs) will be conducted. A comprehensive search strategy will identify relevant studies from MEDLINE, Scopus, CINAHL, SPORTDiscus, CENTRAL and Cochrane Airways Trials Register databases. Studies including adults with and without COPD will be considered. Outcomes will include cardiorespiratory, muscular and cognitive function, intramuscular adaptations, lung volumes and cardiometabolic responses. The protocol is reported according to the Preferred Reporting Items for Systematic Reviews and Meta-Analyses Protocols and the Cochrane Methodological Expectations of Cochrane Intervention Reviews. Risk of bias assessment will be conducted using Cochrane Risk-of-Bias 2 Tool (for RCTs), Risk-of-Bias in Non-Randomised Studies Tool (for NRSIs) and Downs and Black checklist (for CSS). Meta-analyses will be conducted when appropriate, supplemented with a systematic synthesis without meta-analysis.

**Ethics and dissemination** As this study is a systematic review, ethical approval is not required. The final review results will be submitted for publication in a peer-reviewed journal and presented at international conferences.

**PROSPERO registration number** CRD42022307577

## STRENGTHS AND LIMITATIONS OF THIS STUDY

⇒ This protocol is reported according to the Preferred Reporting Items for Systematic Reviews and Meta-Analyses Protocols.
⇒ The protocol includes a comprehensive and peer-reviewed search strategy and broad inclusion criteria to comprehensively synthesise available evidence.
⇒ Exclusion of literature written in languages not known by the research group might exclude relevant literature from the systematic review.

## BACKGROUND

Chronic obstructive pulmonary disease (COPD) is a chronic respiratory disease characterised by persistent and progressive airflow limitation, causing breathlessness, productive coughing, fatigue and recurrent chest infection.[1] COPD is highly prevalent, with the global prevalence among individuals aged ≥40 years being 11.7%.[2] Worldwide, COPD was the third leading cause of death in 2019, according to the WHO,[3] which is expected to increase during the next four decades.[4 5]

Although COPD is primarily a respiratory disease, it is best understood as a systemic disease with several extrapulmonary manifestations.[6 7] Most people (40%–98%[8–10]) with COPD have comorbidities such as cardiovascular disease, diabetes or metabolic syndrome, muscle atrophy, cognitive dysfunction or muscle dysfunction[6 11 12] that directly and substantially impact the disease. On average, a person with COPD suffers from four extrapulmonary manifestations,[13] observed across the entire spectrum of airflow limitation severity.[14] Having one or more comorbidities is associated with more hospitalisations and increased mortality.[15] Nevertheless, extrapulmonary

manifestations are an overlooked aspect of COPD that is not dealt with optimally even though they negatively impact important clinical outcomes independent of the degree of lung impairment.[11 16] For instance, there is an intimate connection between reductions in limb muscle strength and endurance capacity with a reduced quality of life,[17] exercise intolerance,[18–20] greater healthcare utilisation,[21] decreased ability to perform daily activities[22] and increased mortality.[23 24] For example, quadriceps muscle atrophy is associated with a fourfold increase in mortality even after adjusting for age, sex and lung function.[11] At the same time, people with coexisting COPD and cognitive dysfunction have a mortality rate nearly three times as high compared with those having each condition alone.[16]

Similar to the healthy population and most chronic diseases,[25] exercise training is one of the cornerstones in treating extrapulmonary manifestations in COPD, such as decreased muscle, cardiorespiratory or cognitive function.[1 15 26 27] Regular exercise training in people with COPD can increase cardiorespiratory fitness, partly due to increased mitochondrial density and oxidative enzyme activity.[28] Importantly, exercise training has also been shown to reduce dyspnoea and fatigue during daily life activities, decrease anxiety and depression and improve health-related quality of life.[29] It is evident that exercise training improves multiple extrapulmonary manifestations in people with COPD. However, the acute physiological response and the extent of chronic adaptations are repeatedly reported to be altered compared with healthy individuals without COPD and vary among people with COPD.[30] Recently, it was shown that the response to aerobic training concerning mitochondrial function is blunted in COPD compared with matched individuals without COPD.[31] However, a blunted response to exercise training is not a universal finding. For instance, Rabinovich et al[32] reported a difference in oxidative stress between people with COPD and those without COPD following high-intensity training, while Puente-Maestu et al[33] did not. Costes et al,[34] found no change in capillary-to-fibre ratio and mean fibre size in people with COPD after multidisciplinary exercise training, while Gouzi et al[35] showed similar increases in those with and without COPD following endurance training. One study showed that people with COPD have lower mechanical efficiency and exercise capacity in the upper limbs than those without COPD,[36] while others reported preserved capacities.[37] Thus, although multiple studies have been conducted, these seemingly contradictory findings highlight the need for a systematic comparison of studies evaluating acute responses and chronic adaptations to exercise training in people with COPD compared with those without COPD. Increased knowledge about altered and even blunted acute and chronic responses to exercise training in COPD is needed to better tailor exercise training in people with COPD.

## Aims and objectives

This systematic review and meta-analysis aims to synthesise the acute physiological responses and chronic adaptations to exercise training in people with COPD compared with people without COPD. Specifically, the systematic review will address the question: Are there differences in acute physiological responses and chronic adaptations to exercise training in people with or without COPD?

The primary objective is to compare the chronic physiological adaptations to exercise training in people with or without COPD performing the same exercise intervention. The secondary objective is to compare the acute physiological response to exercise training in people with or without COPD performing the same exercise session.

## METHODS

The study protocol of this systematic review is reported in line with the Preferred Reporting Items for Systematic Reviews and Meta-Analyses Protocols (PRISMA-P) guideline (see online supplemental appendix 1)[38] and Cochrane Methodological Expectations of Cochrane Intervention Reviews (MECIR). The systematic review will be reported according to the PRISMA 2020 guidelines,[39] and was registered in the International Prospective Register of Systematic Reviews (PROSPERO) on 27 March 2022 (CRD42022307577).

### Eligibility criteria

Studies will be selected according to the criteria outlined below.

#### Types of study designs

Randomised controlled trials (RCTs), controlled non-randomised studies of interventions (NRSI), cluster trials and prospective comparative cohort studies will be included to synthesise chronic physiological adaptations to exercise training in people with COPD in comparison to people without COPD. Studies presenting a cross-sectional comparison of acute physiological responses to an exercise training session will be included to compare acute physiological responses between people with and without COPD. Retrospective studies, case reports, case series, and opinion pieces will be excluded.

#### Types of participants

Any study including adults ≥18 years of age with a diagnosis of COPD and a group of individuals without COPD (often referred to as 'healthy controls') will be considered. There will be no eligibility criteria for sex or ethnicity or comorbidities in the group of people without COPD. Participants with COPD and comorbidities will not be excluded, but COPD should be the primary disease.

#### Types of intervention

Any type of acute exercise training session and exercise training interventions will be considered as long as data can be separated for and compared between people with COPD and those without COPD. For example, but

not limited to endurance training, interval training, resistance/strength training, neuromuscular electrical stimulation or inspiratory muscle training. For chronic adaptations to exercise training, interventions with a duration of fewer than 3 weeks will be excluded.

## Types of comparator

This systematic review will investigate the differential effects of the same intervention given to different populations, as such, there is no comparator/control group.

## Types of outcome measures

Studies will not be excluded based on outcomes. The following outcomes are of interest, including but not limited to:

1. Cardiorespiratory function
   i. Maximal exercise performance: peak oxygen consumption ($VO_2$peak) and maximal aerobic power during an incremental exercise test.
   ii. Exercise tolerance: constant-work rate cycle test, endurance shuttle walk test, step tests.
2. Cardiorespiratory and metabolic responses during exercise: for example, heart rate, peripheral capillary oxygen saturation, oxygen consumption, carbon dioxide production, respiratory exchange ratio, the concentration of blood lactate, markers of mitochondrial biogenesis, myogenesis, neuroprotection, inflammation, and oxidative stress.
3. Muscle function:
   i. Muscle strength: maximal muscle force (eg, maximal voluntary contraction), isokinetic or isometric (peak torque) and isotonic strength (1-repetition maximum).
   ii. Muscle power: repeated maximal contractions at a submaximal load (ie, to generate an increase in contractile force over a certain unit of time).
   iii. Muscle endurance: change in muscle force over time (eg, time to failure when maximal or submaximal static or dynamic contractions are performed).
4. Functional performance: for example, 6-minute-walk-test, 5-time sit-to-stand test, 30 s sit-to-stand and 1 min sit-to-stand test.
5. Intramuscular adaptations: for example, mitochondrial function and markers, markers of oxidative stress and inflammation, enzyme activity, fibre type proportion, fibre size and capillarizsation.
6. Lung volumes: for example, tidal volume, minute ventilation, breathing frequency, ventilation to maximum voluntary ventilation ratio, dynamic hyperinflation, end-expiratory lung volume, end-inspiratory lung volume, and inspiratory capacity.
7. Cognitive function: global cognitive tests (eg, Montreal Cognitive Assessment, Mini-Mental State Examination, General Practitioner Assessment of Cognition, Mini-Cog Test).

## Timing

Only before and immediately after intervention data will be included for intervention studies with more than two time points of assessments. If an intervention study measures acute responses before and after the intervention, only before (baseline) measurements will be included.

## Setting

There will be no exclusion by type of setting.

## Timespan

All articles from inception to the start of the literature search will be considered. Any restriction of the time period is not deemed relevant.

## Language

English, Swedish, Norwegian, Danish, French, and Dutch literature will be considered. Articles written in other languages will be excluded.

## SEARCH METHODS FOR IDENTIFICATION OF STUDIES
### Electronic searches

Relevant studies will be identified from (database inception to date):

1. MEDLINE (Ovid SP interface)
2. Scopus
3. CINAHL and SPORTDiscus (EBSCO interface)
4. Cochrane Central Register of Controlled Trials (CENTRAL)
5. Cochrane Airways Trials Register

Search for grey literature, registered trials and ongoing studies will be performed, from the inception of the database to date, through but not limited to:

1. WHO International Clinical Trials Registry Platform (ICTRP)
2. International Standard Randomised Controlled Trial Number (ISRCTN)
3. National Institutes of Health Clinical Trials Database (ClinicalTrials.gov)

Corresponding authors or principal investigators of identified unpublished and studies in progress will be contacted to establish whether published literature was missed.

### Other resources

Relevant conference proceedings will be searched, and if required, we will contact authors of identified trials for unpublished studies or missing information. If sufficient data are provided, conference abstracts will be included.

The reference list of included studies identified through the search will be hand-searched for additional studies. Other systematic reviews on the topic will also be checked for further studies. The 'related articles' function in PubMed will be used on included studies or relevant reviews.

The Cochrane Database of Systematic Reviews and PROSPERO have also been searched for existing or ongoing reviews on the topic. No previous or ongoing systematic review was identified. We will search PubMed for errata or retractions from included studies published in full text and report which date this was done within the review.

## Search strategy

The search strategy is based on the Cochrane Airways Group Register of Trials guidelines. It has been developed with the assistance of a health science librarian and is peer-reviewed by experts in exercise training, pulmonary rehabilitation and lung diseases. To ensure literature saturation, comprehensive searches will be constructed of both index terming (Medical subject headings (MeSH) terms), free text terms and synonyms. No language restriction will be imposed on the search. A draft of the search strategy for MEDLINE (OVID interface) is presented in the online supplemental appendix 2. The final MEDLINE search strategy will be adapted to other databases' and trial registries' syntax and subject headings.

## Data collection and analysis

### Study records

The results from the literature searches will be uploaded to Covidence (Covidence, Melbourne, Australia). This internet-based software programme facilitates the management of studies, including removing duplicates and collaboration among reviewers during the study selection process. The research team will develop, test and refine screening questions and forms for assessments based on the eligibility criteria. Members of the review team not familiar with Covidence will be trained in the software before the start of the review. The search process will be documented, including:

1. the name of the database searched;
2. the name of the database provider/system used
3. the date when the search was run
4. the years covered by the search;
5. the search terms used, hits per search term and number of articles retrieved.

### Selection process

Two review authors (JJ and JDB) will screen the titles and abstracts of the search results independently and code them as 'retrieve' (eligible or potentially eligible/unclear) or 'do not retrieve'. Then, we will retrieve the full-text study reports of all potentially eligible studies. Two review authors (JJ and JDB) will independently screen them for inclusion, recording the reasons for exclusion of ineligible studies. If needed, additional information from study authors will be sought using e-mail to resolve questions about eligibility. We will resolve any disagreement through discussion or if required, we will consult a third review author (AN), and a majority rule will be used. Neither of the review authors will be blind to the journal titles, the study authors or institutions. Special attention will be given to identifying and excluding duplicates and collating multiple reports of the same study. Rather than each report, each study is the unit of interest in the review. We will record the selection process in sufficient detail to complete a PRISMA flowchart diagram. The inter-assessor agreement will be assessed using Kappa statistics.

## Data collection process

Two reviewers (JJ and JDB) will use a standardised data extraction form to extract data independently from full texts of all included studies. The form will be pilot-tested on two to three potentially eligible articles and reviewers trained in using the form. Disagreement will be solved by consensus. When disagreement cannot be resolved by consensus, a third reviewer (AN) will be consulted, and a majority rule will be used. In the absence of complete descriptions of outcomes and effect estimates, authors will be asked for additional information or missing data using e-mail. All data will be double-checked with the included studies by a third reviewer (AN). The following study characteristics will be extracted:

1. Study: author(s) name, title, publication year.
2. Methods: study design, total duration of the study, number of study centres (if multicentre), study setting
3. Participants: sample size, mean age, age range, sex, inclusion and exclusion criteria, baseline lung function and the number of comorbidities. For COPD group: severity of the condition.
4. Intervention: type of exercise training, intensity, duration, frequency and progression principles
5. Outcomes: data will be sought on the following outcomes—maximal exercise performance, exercise capacity, cardiorespiratory and metabolic responses during exercise, muscle strength, muscle power, muscle endurance, functional performance, intramuscular adaptations and lung volumes. Chronic adaptation following an exercise intervention period in maximal exercise performance, exercise tolerance and muscle function are the primary outcomes in this systematic review since these are vital objectives in international COPD treatment guidelines.[6 40 41] Other chronic adaptations and acute responses from a single exercise session in the abovementioned outcomes are secondary outcomes. The 'Type of outcome measures' section includes detailed descriptions of selected outcomes.

One review author (JJ) will transfer data into Review Manager software (Revman, Cochrane Collaboration, Oxford, UK). We will double-check that data are entered correctly by comparing the data presented in the systematic review with the study reports. A second review author (JDB) will spot-check study characteristics for accuracy against the study report.

## Risk of bias assessment

The risk of bias (RoB) in eligible studies will be assessed independently by two authors (JJ and JDB) using appropriate tools. We will resolve any disagreements by discussing or involving another author (AN). No study will be excluded due to poor quality, but sensitivity analyses excluding poor-quality studies will be performed.

RCTs will be assessed using Risk of Bias Tool 2 (RoB 2).[42 43] We will assess the RoB in the following five domains:

1. bias arising from the randomisation process;
2. bias due to deviations from intended interventions;
3. bias due to missing outcome data;

4. bias in measurement of the outcome; and

5. bias in selection of the reported result.

Signalling questions and the algorithm in the RoB 2 Tool will provide the basis for the judgement about the RoB. If there is sufficient reason to override the algorithm, we will do this and state a reason for it. To manage the assessment of bias of included studies we will use templates and spreadsheets of the RoB 2 Tool.[43] An overall RoB for each result will be produced, based on the least favourable assessment of bias across the domains according to:

► low RoB;

► some concerns; or

► high RoB.

The Risk of Bias in Non-randomised Studies of Interventions (ROBINS-I)[44] will be used for assessing the quality of NRSIs. This is the recommended tool by the Cochrane Scientific Committee to assess the RoB in NRSIs. The ROBINS-I tool is similar to the RoB 2 Tool and comprises a series of signalling questions assessing the bias in seven domains:

1. bias due to confounding;

2. bias in selection of participants for the study;

3. bias in classification of interventions;

4. bias due to deviations from intended interventions;

5. bias due to missing data;

6. bias in measurement of outcomes; and

7. bias in selection of the reported result.

Important confounding domains that are prognostic factors for the outcomes of interest are the extent of matching between the groups with and without COPD, especially regarding daily physical activity (PA) and any differences in cardiorespiratory fitness.

We will reach a RoB judgement and assign one of the five levels to each domain and an overall judgement per study:

1. low RoB;

2. moderate RoB;

3. serious RoB;

4. critical RoB; or

5. no information (on which to base a judgement about RoB).

For cross-sectional studies (CSSs), the RoB will be assessed using a modified version of the checklist by Downs and Black.[45] This checklist has been recommended for assessing the quality of RCT and NRSI, as well as CSSs.[46] The 14 most relevant items will be used. The 14 items are appropriate for NRSI and have previously been used in studies within the field.[47] The psychometric properties of the checklist have been validated and reported elsewhere.[45] The 14 items, previously described[47] are yes (1 point) or no (0 points) questions resulting in a score between 0 and 14, where higher scores indicate higher methodological quality. For subsequent sensitivity analysis, the following cut-points, based on previously used cut-offs for the original list,[48] will be used to categorise studies by quality: 'Excellent' (13–14 points), 'good' (10–12 points), 'fair' (8–9 points) or 'poor' ≤7 points).

The 14 items checklist can be found in online supplemental appendix 3.

## Synthesis

Data will be meta-analysed in RevMan and will be presented in forest plots and a table providing a summary of study characteristics and the findings. Other computer software, such as Jamovi might also be used for supplemental analyses. For data that cannot be meta-analysed, that is, due to a lack of data to calculate effect sizes or too diverse study characteristics, a systematic synthesis without meta-analysis (SWiM)[49] will be presented in the text and with graphics. The SWiM will be conducted according to the PRISMA and SWiM[49] reporting guidelines. The SWiM guideline is a nine-item checklist to promote transparent reporting of systematic reviews.[49]

Meta-analyses will be undertaken where meaningful; if the participants, type of intervention (ie, resistance training, continuous training, high-intensity interval training) and study design (ie, acute or chronic) are homogenous enough to warrant pooling. We anticipate only continuous data to be reported for the outcomes of interest. Continuous data will be analysed as the mean difference (MD), where outcome data are reported on a uniform scale or standardised mean difference (SMD) when different metrics are used but deemed clinically homogenous. Meta-analyses will be based on change from baseline rather than postintervention values when available. Data extraction (and any needed imputations and calculations) will be done according to recommendations in *Cochrane Handbook for Systematic Reviews of Interventions* V.6.2, section 6.5.2.[42] If needed, additional information from the study authors will be sought using e-mail. A random-effects model will be used to allow for random error and inter-study variability.

To compare the physiological effects of exercise, we will use per-protocol analyses as the effect of interest, where they are reported. If only intention-to-treat data are reported, this will be used for analysis. If both per-protocol and intention-to-treat data are reported, the intention-to-treat data will be used in a sensitivity analysis. Where multiple trial arms are reported in a single study, we will include only relevant ones. If applicable, we will combine intervention arms or reported subgroups as per the recommendations in *Cochrane Handbook for Systematic Reviews of Interventions* V.6.2,[42] sections 6.2 and 6.5, respectively.

## Assessment of heterogeneity

We will use the $I^2$ and $\tau^2$ statistics to measure heterogeneity between the studies in each analysis. If we identify substantial heterogeneity ($I^2 \geq 50\%$), we will report it and explore the possible causes by prespecified subgroup analysis. The importance of the $I^2$ statistic will be interpreted together with the CI and p value from the $\chi^2$ statistic.

## Assessment of reporting biases

If we can pool more than 10 studies, we will create and examine the asymmetry of a funnel plot and perform Egger's test to explore possible small study and publication biases. The possible influence of small-study/publication biases on review findings will be a part of our 'Risk of bias' assessment and Grading of Recommendations Assessment, Development, and Evaluation (GRADE) assessments of the quality of the evidence.

## Subgroup analysis and investigation of heterogeneity

We plan to carry out the following subgroup analyses if adequate data can be obtained:

1. the extent of matching on PA (matched with any type of method compared with not matched);
2. duration of intervention ≥15 sessions compared with ≤16 sessions;
3. severity of the condition in participants with COPD (GOLD assessment[1] A and B vs C and D);
4. sex;
5. exercise intensity; and
6. body mass index ≥21 compared with <21 (a cut-off value often used in COPD[11] for underweight).

Relevant subgroup analyses may be extended to random-effect meta-regression analyses if an adequate number of studies include a specific outcome. For example, this could be subgroup analysis 1 with more than two categorical variables (ie, no matching, subjectively measured PA and objectively measured PA) or subgroup analysis 2 with the duration of intervention as a continuous explanatory variable. The outcome variable will be the effect estimate, that is, MD or SMD.

## Sensitivity analysis

We plan to carry out the following sensitivity analyses, removing the following from the primary outcome analyses:

1. Trials judged in the quality assessment to be at high risk of bias (RoB 2), moderate or critical risk of bias (ROBINS-I) or poor quality (Downs and Black).
2. Per-protocol analyses (either removing the whole trial or using data for intention-to-treat if this is reported too).

We will compare results from a fixed-effect model versus a random-effect model. If we impute SD using correlation coefficients in the synthesis, sensitivity analyses will be performed with different values of the correlation coefficient to determine if the overall results are robust to the imputed correlation coefficients.[42] When appropriate, we will compare the results from an absolute change from baseline versus a percentage change from baseline meta-analysis.

The sensitivity analyses will be presented in a summary table. Other suitable sensitivity analyses, as identified during the review process, may also be performed.

## Assessment of the certainty of the evidence

We will use the GRADE approach to assess the quality of a body of evidence related to the outcomes. Thus, the quality of the evidence will be graded as 'high', 'moderate', 'low' or 'very low'. We will use the methods and recommendations described in the *Cochrane Handbook for Systematic Reviews of Interventions* V.6.2[42] using GRADEpro software.

## Patient and public involvement

Patients and/or the public were not involved in the design, conduct, reporting, or dissemination plans of this research.

## DISCUSSION

Registration and publication of this protocol will promote and maintain transparency in the review process and minimise the RoBs and counteract duplication of similar reviews. In addition, a high methodological standard will be supported by adhering to PRISMA-P, SWiM and MECIR guidelines. A potential limitation of this systematic review is the exclusion of literature written in languages not known by the research group might leave relevant literature out of the systematic review.

## ETHICS AND DISSEMINATION

As this study is a systematic review, ethical approval is not required. The systematic review will be submitted for publication in a peer-reviewed journal. We also intend to present the findings at national and international conferences and inform patient organisations. There has been no patient or public involvement in the design of the protocol or dissemination plans.

## Amendments

We will conduct the review according to this protocol and justify any deviations in a specific section of the systematic review.

**Contributors** AN conceptualised the study as the guarantor for the protocol. JJ and AN contributed to developing the eligibility criteria, the search strategy, the risk of bias assessment, data extraction and data synthesis. JJ wrote the initial protocol, while AN and JDB provided critical insight and revisions to the manuscript. All authors read, approved and contributed to the final written manuscript.

**Funding** Funding This review is being undertaken as part of JJ's PhD work, funded by the Swedish Research Council (Vetenskapsrådet), grant number: 2020-01296.

**Competing interests** None declared.

**Patient and public involvement** Patients and/or the public were not involved in the design, or conduct, or reporting or dissemination plans of this research.

**Patient consent for publication** Not applicable.

**Provenance and peer review** Not commissioned; externally peer reviewed.

**ORCID iDs**
Johan Jakobsson http://orcid.org/0000-0002-9816-194X
Jana De Brandt http://orcid.org/0000-0003-3463-1911
André Nyberg http://orcid.org/0000-0003-2782-7959

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
