## [Reviewer comments · BMJ Open]

ARTICLE DETAILS

TITLE (PROVISIONAL)	Physiological Responses and Adaptations to Exercise Training in People with or without Chronic Obstructive Pulmonary Disease: Protocol for a Systematic Review and Meta-Analysis
AUTHORS	Jakobsson, Johan; De Brandt, Jana; Nyberg, André

VERSION 1 – REVIEW

REVIEWER	Knut Sindre Mølmen Inland Norway University of Applied Sciences, Section for Health and Exercise Physiology
REVIEW RETURNED	05-Jul-2022

GENERAL COMMENTS	In this manuscript, Jakobssen and colleagues are outlining a systematic review study aiming "to synthesize and compare the physiological acute responses and chronic adaptations to exercise training in people with COPD compared to people without COPD". The manuscript is well-written, and in my opinion, it is highlighting a very interesting topic regarding exercise training responsiveness in COPD subjects. The manuscript also addresses a possibly very overlooked topic, i.e. physical activity/exercise training as therapy for COPD subjects. However, I have some concerns/comments that needs to be addressed. 1. "Change from baseline will be the preferred metric for outcomes". Please elaborate on why you will use absolute change scores and not relative changes for comparing COPD and healthy responses to exercise. As COPD subjects typically have suppressed levels at baseline compared to healthy individuals for the variables you want to investigate, the interpretation from delta and % delta will most likely differ (as seen in e.g. Mølmen, K.S., Hammarström, D., Falch, G.S. et al. Chronic obstructive pulmonary disease does not impair responses to resistance training. J Transl Med 19, 292 (2021). https://doi.org/10.1186/s12967-021-02969-1).2. I'm a bit concerned that there are too few studies investigating COPD vs healthy training responses for many of the variables you are aiming to study. E.g. regarding adaptations to resistance training, I am only familiar with two different study interventions investigating this: 1) Constantin et al., 2013 (Thorax) / Menon et al., 2012 (Respir Res) / Menon et al., 2012 (Chest) and 2) Mølmen et al., 2021 (J Transl Med) / Oberholzer & Mølmen et al., 2022 (JCSM Rapid Com). Probably there are more studies regarding endurance training, but not an overwhelming quantity there either. Will there be a sufficient amount of studies to conduct meta-analyses, or do you rather have to systematically review the available data for most of the outcome variables?
---

	3. In extension of question 2. You write "Meta-analyses will be undertaken where meaningful; if the participants, intervention, and study design are similar enough for pooling to make sense". Please elaborate and specify on what kind of interventions that can be pooled for meta-analysis purposes. As for example the senior author of the current manuscript have successfully demonstrated earlier, different training modalities/exercises can to a varying degree lead to different acute responses to exercise between COPD and healthy subjects. I.e. COPD subjects are more performance-limited by their low cardiorespiratory capacity in exercises posing large whole-body metabolic demands. This may imply that some training interventions are more favorable for inducing long-term adaptations than others for COPD subjects, and that there should be specified what kind of interventions that will be pooled for meta-analysis purposes in the current study. Maybe it is appropriate to divide studies into the exercise training categories continuous endurance exercise / high-intensity endurance exercise (interval training?) / resistance exercise? 4. Please prioritize and specify your outcome variables to a greater degree, and in accordance with the Cochrane Handbook, https://handbook-5-1.cochrane.org/chapter_5/5_4_2_prioritizing_outcomes_main_primary_and_secondary_outcomes.htm 5. What kind of RCTs are you planning to include? The independent variable (COPD/healthy) of interest in the study is not possible to manipulate. 6. Please highlight why you will do intention-to-treat rather than per-protocol analyses. The primary objective of the current study is to compare the chronic physiological adaptations to exercise training in people with or without COPD performing the same exercise intervention. In my opinion, it is crucial that the study participants have actually conducted the protocol if you want to elucidate on this primary objective.
--	--

REVIEWER	Alex Jenkins McGill University, Clinical Exercise & Respiratory Physiolo
REVIEW RETURNED	11-Jul-2022

GENERAL COMMENTS	Thank you for allowing me the opportunity to review this protocol which will assess the acute and chronic physiological responses to exercise in people with or without COPD. This review includes important, and often overlooked outcomes which will add to our understanding of physiological as well as immunological responses to exercise in COPD when compared to 'healthy' populations. I'd like to thank the authors for pursuing this work covering a significant amount of literature as evidenced by identifying >6000 articles from the search. This protocol is very well written and follows robust methodological criteria. I was particularly pleased to see that study interventions will be characterised in line with the FITT principles. It was a pleasure to review, and I look forward to reading the final published work! I only have minor comments/suggestions to add.  - Abstract  o Lines 31-33: This seems a little vague, I might suggest to change this to state that MECIR guidelines were followed or something similar? - Background  o Pg 4, Line 6: remove 'a' before 'persistent'
--

	 o Pg 4, Line 10: clarify that these statistics are based on worldwide data o Aims and objectives line 15: change 'aim' to 'aims' - Methods o Pg 5, Lines 33-34: I'd suggest either removing the date (not necessary to include in my opinion) or amending the sentence as it jumps between present and past tense o Types of study designs: How will you deal with potential studies that have assessed acute physiological responses pre- and post-intervention (i.e., pre and post at baseline vs pre and post at end of intervention)? Will these chronic 'acute' changes be captured in the chronic adaptations? o I would make it clear that there is no comparator/control but that you are interested in a population comparison to the same intervention as done in your PROSPERO protocol o Types of participants: 'Should be confirmed by spirometry...' Should be quite vague to use as an inclusion/exclusion term and this form of diagnosis is often not reported in some cases unfortunately. Authors often report physician confirmed diagnosis with no mention of confirmatory spirometry. I would suggest amending this to allow for a bit of flexibility in your protocol. o Types of outcome measures: There are a lot of outcomes included in this review, and it's more a pre-emptive thought, but it will be worthwhile considering how data unable to be included in meta-analyses can be presented in a 'reader-friendly' way whilst also conforming to SWiM guidance. o Timing: By 'after intervention' do you mean immediately after? If so, please state. o Language: Will any attempts be made to translate articles falling outside of these languages? If not, explicitly state these will be excluded straight away. If so, state that attempts will be made to translate articles. o Was there a reason for not including EMBASE as a search database? o Pg 7, line 28: Delete space between 'data' and 'base' o Pg 9, lines 27-29: I'd suggest deleting the sentence 'Change from baseline will be the preferred metric for outcomes' as this is better placed in the data synthesis section where you have discussed in more detail o Pg 10, Lines 54-57: Are these cut-points arbitrary or supported by previous evidence? Please state. o Pg 11, Line 44: suggest rewording 'similar enough for pooling to make sense' to 'homogeneous enough to warrant pooling'. o Synthesis: How will you deal with studies that do not report mean differences? What is your plan in terms of hierarchy? Will you seek to obtain data in the correct format from authors first? If that fails, will you look to calculate a correlation coefficient for the SD of change based on data from another study? If that fails as well, will you then look to apply an arbitrary correlation coefficient of 0.5 and then perform sensitivity analyses? Or will you just use post-intervention only values if baseline values are deemed similar enough? It would be worthwhile stating a little more detail for this. o Pg 12, Lines 3-4: You may have instances where there are more than one relevant intervention arm. I would suggest being open to combining groups as per Cochrane guidance. Of course, this should be used sparingly, and if one group is the most relevant, then focus on this group. I suggest adding a sentence in relation to how you will approach these studies. o Pg 12, heterogeneity: What is the prespecified threshold used for suggesting substantial heterogeneity to warrant subgroup analyses?
--	--

	o Pg 12, subgroup analyses: What is the rationale for 20 sessions as a cut-off for intervention duration? I just wondered that with pulmonary rehabilitation being 12-16 sessions long (usually!) that this maybe a more suitable way of guiding the cut-off (i.e., >16 vs ≤16)? o Pg 12, subgroup analyses: Given your outcomes can be significantly influenced by body mass, would it be possible to include this as a prespecified subgroup? Especially given that sex is also included as a subgroup. Maybe BMI, body mass/weight and/or a measure of fat-free mass could be used? o Pg 12, sensitivity analyses: In relation to my earlier comment about correlation coefficients, it will be important to perform sensitivity analyses to make sure these do not significantly impact the pooled estimates. So, I would recommend stating that sensitivity analyses will be performed for studies where the SD of change has been imputed using a correlation coefficient.
--	--

REVIEWER	Rui Vilarinho Polytechnic Institute of Porto
REVIEW RETURNED	12-Jul-2022

GENERAL COMMENTS	Thank you for asking me to review this manuscript presenting a protocol for a systematic review and meta-analysis of physiological responses and adaptations to exercise training in people with or without chronic obstructive pulmonary disease. As strengths of this manuscript, this protocol will be reported according to the Preferred Reporting Items for Systematic Reviews and Meta-Analyses Protocols, is registered at PROSPERO, and the risk of bias assessment is mentioned. During comments and suggestions, please consider the page number presented upper left corner. General comment: Throughout the manuscript, several terms are used to describe the same information (e.g, "physical capacity", "cardiorespiratory fitness", "physical capacity"). Although there are similarities between them, these terms refer to different definitions. Can authors explain the reasons for using several terms and try to uniformize the information? KEYWORDS 1) The keyword "exercise training" is duplicated (Title and Keywords). Can the authors include new keywords and different from the title to enhance the search in the future publication of the protocol? INTRODUCTION 2) Page 5, line 15. Reference 6 may not be sufficient for the information that is presented. Please consider including more pertinent references such as: Machado A, Marques A, Burtin C. Extra-pulmonary manifestations of COPD and the role of pulmonary rehabilitation: a symptom-centered approach. Expert Rev Respir Med. 2021 doi: 10.1080/17476348.2021.1854737. 3) According to the studies cited in the Introduction section (lines 52-60), it seems that a variety of exercise interventions could be an important barrier/limitation for this systematic review meta-analysis.
---

	The authors of this manuscript presented studies that applied "high-intensity training" (reference 31), "multidisciplinary exercise training" (reference 3), and "endurance training" (reference 34). Do authors believe that this information is important to be included in the manuscript as a possible limitation? METHODS 4) Throughout the manuscript, "healthy controls" or "people without COPD" are mentioned to present the same population. Can the authors standardize the designation of this population in a single term? However, it is not guaranteed that "healthy controls" and "people without COPD" have the same characteristics. Do authors believe that is important to certify in the studies that will be selected for this systematic review the type and number of comorbidities of "people without COPD" (e.g, no statistical differences in the number of comorbidities or no differences in comorbidity indexes between groups)? Can this information influence the objectives and the further results of the meta-analysis? 5) Please clarify how the diagnosis of COPD based solely on spirometry would not misclassify chronic persistent asthma as COPD. 6) How the outcome measures will be selected according to the interventions. For example, it will be used the same outcome measures for neuromuscular electrical stimulation (NMES) and inspiratory muscle training (IMT)? 7) Page 7 line 13-14. Can the authors include a reference to support this information? 8) In outcomes measures: can respiratory exchange ratio (RER) be included? Furthermore, do authors consider that it is important to select the outcomes measures according to the level of evidence on their measurement properties (according to COSMIN and GRADE), namely for the tests mentioned in "exercise capacity", "functional performance", "cognitive function"? 9) Page 7, in Types of outcome measures subsection, it is important to clarify the definitions of exercise capacity and maximal exercise performance. They are not the same? Can the 6MWT and step tests be considered more appropriate as physical tests to assess functional capacity? 10) Page 13. Subgroup analysis and investigation of heterogeneity subsection: reasons for "duration of intervention \geq 20 sessions compared to $<$ 20 sessions"? REFERENCES 11) Errors were found in some references. For example, reference 34.
--	--

VERSION 1 – AUTHOR RESPONSE

Reviewer: 1

Dr. Knut Sindre Mølmen, Inland Norway University of Applied Sciences

Comments to the Author:

In this manuscript, Jakobssen and colleagues are outlining a systematic review study aiming "to synthesize and compare the physiological acute responses and chronic adaptations to exercise training in people with COPD compared to people without COPD". The manuscript is well-written, and in my opinion, it is highlighting a very interesting topic regarding exercise training responsiveness in COPD subjects. The manuscript also addresses a possibly very overlooked topic, i.e. physical activity/exercise training as therapy for COPD subjects. However, I have some concerns/comments that needs to be addressed.

Thank you for the thorough review, kind words and helpful comments to the manuscript.

1. "Change from baseline will be the preferred metric for outcomes". Please elaborate on why you will use absolute change scores and not relative changes for comparing COPD and healthy responses to exercise. As COPD subjects typically have suppressed levels at baseline compared to healthy individuals for the variables you want to investigate, the interpretation from delta and % delta will most likely differ (as seen in e.g. Mølmen, K.S., Hammarström, D., Falch, G.S. et al. Chronic obstructive pulmonary disease does not impair responses to resistance training. *J Transl Med* 19, 292 (2021). <https://doi.org/10.1186/s12967-021-02969-1>).

Thank you for bringing up this excellent comment. We had not considered this in detail (absolute versus percentage) and agree with your comments to large extent. However, we still aim to have absolute change as the primary method of analysis. The rationale for this is that it is the mainly recommended method as per Cochrane guidelines, and MCIDs are most often in absolute values. Using the percentage change might also be less statistically robust (<https://bmcmedresmethodol.biomedcentral.com/articles/10.1186/1471-2288-1-6#citeas>) and be misleading in some type of measures. However, if appropriate, we will perform a percentage change analysis as a sensitivity analysis. (page 13, line 7-9).

2. I'm a bit concerned that there are too few studies investigating COPD vs healthy training responses for many of the variables you are aiming to study. E.g. regarding adaptations to resistance training, I am only familiar with two different study interventions investigating this: 1) Constantin et al., 2013 (*Thorax*) / Menon et al., 2012 (*Respir Res*) / Menon et al., 2012 (*Chest*) and 2) Mølmen et al., 2021 (*J Transl Med*) / Oberholzer & Mølmen et al., 2022 (*JCSM Rapid Com*). Probably there are more studies regarding endurance training, but not an overwhelming quantity there either. Will there be a sufficient amount of studies to conduct meta-analyses, or do you rather have to systematically review the available data for most of the outcome variables?

We are aware that for many outcomes there will be too few studies to meta-analyse. However, we will still do a comprehensive search, not excluding outcomes on this basis. This will provide knowledge on the potential scarcity of some types of studies. We anticipate there will be sufficient amount of studies to conduct meta-analyses for some outcomes, while we acknowledge we will need to systematically review the data for many outcomes narratively, or conclude that there is no or limited amount of data for some outcomes.

3. In extension of question 2. You write "Meta-analyses will be undertaken where meaningful; if the participants, intervention, and study design are similar enough for pooling to make sense". Please elaborate and specify on what kind of interventions that can be pooled for meta-analysis purposes. As for example the senior author of the current manuscript have successfully demonstrated earlier, different training modalities/exercises can to a varying degree lead to different acute responses to exercise between COPD and healthy subjects. I.e. COPD subjects are more performance-limited by their low cardiorespiratory capacity in exercises posing large whole-body metabolic demands. This may imply that some training interventions are more favourable for inducing long-term adaptations than others for COPD subjects, and that there should be specified what kind of interventions that will be pooled for meta-analysis purposes in the current study. Maybe it is appropriate to divide studies

into the exercise training categories continuous endurance exercise / high-intensity endurance exercise (interval training?) / resistance exercise?

We agree with you that it would be appropriate to divide studies if there are enough studies to separate on intervention. We have provided some clarification on this. However, if there are too few studies to separate between, for example, continuous and interval-training (both endurance based), we think it still is valuable to do the meta-analysis, while acknowledging the limitation. If possible, this will be further analysed with sub-group analyses (exercise intensity, i.e. low/moderate intensity versus high intensity (interval) training. (Page 11, line 38-40).

4. Please prioritize and specify your outcome variables to a greater degree, and in accordance with the [Cochrane Handbook, https://handbook-5-1.cochrane.org/chapter_5/5_4_2_prioritizing_outcomes_main_primary_and_secondary_outcomes.htm](https://handbook-5-1.cochrane.org/chapter_5/5_4_2_prioritizing_outcomes_main_primary_and_secondary_outcomes.htm)

The primary outcomes have been limited and more specified:

“Chronic adaptation following an exercise intervention period in maximal exercise performance, exercise tolerance and muscle function are the primary outcomes in this systematic review since these are vital objectives in international COPD treatment guidelines ¹⁻³. Other chronic adaptations and acute responses from a single exercise session in the abovementioned outcomes are secondary outcomes.”

(Page 9, line 21-25)

5. What kind of RCTs are you planning to include? The independent variable (COPD/healthy) of interest in the study is not possible to manipulate.

This could for example be a study with a group of people with COPD and controls, of which both are randomized into two or more types of intervention arms. But, it can also be non-RCTs where the differential effects of the same intervention given to different populations are compared.

6. Please highlight why you will do intention-to-treat rather than per-protocol analyses. The primary objective of the current study is to compare the chronic physiological adaptations to exercise training in people with or without COPD performing the same exercise intervention. In my opinion, it is crucial that the study participants have actually conducted the protocol if you want to elucidate on this primary objective.

This was an excellent comment. While we initially adhered to Cochrane guidance, doing intention-to-treat, we agree that per-protocol is in many cases more appropriate. Thus, we will do per-protocol as a main analysis while also doing a sensitivity analysis using ITT, if this is feasible depending on the reported data.

Pg 12, line 6-10: *“To compare the physiological effects of exercise, we will use per-protocol analyses as the effect of interest, where they are reported. If only intention-to-treat data is reported, this will be used for analysis. If both per-protocol and intention-to-treat data is reported, the intention-to-treat data will be used in a sensitivity analysis.”*

Reviewer: 2

Dr. Alex Jenkins, McGill University

Comments to the Author:

Thank you for allowing me the opportunity to review this protocol which will assess the acute and chronic physiological responses to exercise in people with or without COPD. This review includes important, and often overlooked outcomes which will add to our understanding of physiological as well

as immunological responses to exercise in COPD when compared to 'healthy' populations. I'd like to thank the authors for pursuing this work covering a significant amount of literature as evidenced by identifying >6000 articles from the search. This protocol is very well written and follows robust methodological criteria. I was particularly pleased to see that study interventions will be characterised in line with the FITT principles. It was a pleasure to review, and I look forward to reading the final published work! I only have minor comments/suggestions to add.

Thank you for reviewing this manuscript and providing helpful comments that has improved the manuscript.

Abstract

Lines 31-33: This seems a little vague, I might suggest to change this to state that MECIR guidelines were followed or something similar?

We agree, this has been changed.

"The protocol is reported according to the Preferred Reporting Items for Systematic Reviews and Meta-Analyses Protocols and the Cochrane Methodological Expectations of Cochrane Intervention Reviews."

Background

Pg 4, Line 6: remove 'a' before 'persistent'

Pg 4, Line 10: clarify that these statistics are based on worldwide data

Aims and objectives line 15: change 'aim' to 'aims'

These adjustments have been made.

Methods

o Pg 5, Lines 33-34: I'd suggest either removing the date (not necessary to include in my opinion) or amending the sentence as it jumps between present and past tense

We have amended the sentence to past tense only.

Types of study designs: How will you deal with potential studies that have assessed acute physiological responses pre- and post-intervention (i.e., pre and post at baseline vs pre and post at end of intervention)? Will these chronic 'acute' changes be captured in the chronic adaptations?

This is a good question that we did not consider. We have clarified that, in this case, we will use only the pre-intervention data for acute responses.

Page 7, line 1-4: *"If an intervention study measures acute responses before and after the intervention, only before (baseline) measurements will be included."*

I would make it clear that there is no comparator/control but that you are interested in a population comparison to the same intervention as done in your PROSPERO protocol

This is a good suggestion that we have implemented.

Pg 6, line 8-10: *"Types of comparator*

This systematic review will investigate the differential effects of the same intervention given to different populations, as such, there is no comparator/control group. "

Types of participants: 'Should be confirmed by spirometry...' Should is quite vague to use as an inclusion/exclusion term and this form of diagnosis is often not reported in some cases unfortunately. Authors often report physician confirmed diagnosis with no mention of confirmatory spirometry. I would suggest amending this to allow for a bit of flexibility in your protocol.

This is a good thought, and we have, after some consideration, changed this accordingly to “with a diagnosis of COPD”. Pg 5 line 36.

Types of outcome measures: There are a lot of outcomes included in this review, and it’s more a pre-emptive thought, but it will be worthwhile considering how data unable to be included in meta-analyses can be presented in a ‘reader-friendly’ way whilst also conforming to SWiM guidance.

This is a good thought. We will continue to consider this and the final result will be depending on how much data/papers there are to consider for the review.

Timing: By ‘after intervention’ do you mean immediately after? If so, please state.

That is correct, this has been clarified.

“If an intervention study measures acute responses before and after the intervention, only before (baseline) measurements will be included.”

Language: Will any attempts be made to translate articles falling outside of these languages? If not, explicitly state these will be excluded straight away. If so, state that attempts will be made to translate articles.

The plan is not to translate articles falling outside of these languages, this has now been clarified.

*“Language
English, Swedish, Norwegian, Danish, French, and Dutch literature will be considered. Articles written in other languages will be excluded.”*

Was there a reason for not including EMBASE as a search database?

It was initially planned to be included, but this is solely due to the fact that our department do not have a subscription/access to it at this time. While perhaps not optimal, we reason (as do our librarian) that we will have a very high coverage with Medline, CENTRAL, CINAHL, SPORTDiscus and CAGR.

Pg 7, line 28: Delete space between ‘data’ and ‘base’

The space has been deleted.

Pg 9, lines 27-29: I’d suggest deleting the sentence ‘Change from baseline will be the preferred metric for outcomes’ as this is better placed in the data synthesis section where you have discussed in more detail

We agree and have deleted the sentence.

Pg 10, Lines 54-57: Are these cut-points arbitrary or supported by previous evidence? Please state.

The checklist does not have these cut-points originally. But for the sake of sensitivity analyses, removing poor-quality studies, we need cut-points. These cut-points are based on previous suggested cut-points (as now clarified) for the full checklist.

Pg 11, Line 44: suggest rewording ‘similar enough for pooling to make sense’ to ‘homogeneous enough to warrant pooling’.

This sentence has been rephrased.

Synthesis: How will you deal with studies that do not report mean differences? What is your plan in terms of hierarchy? Will you seek to obtain data in the correct format from authors first? If that fails, will you look to calculate a correlation coefficient for the SD of change based on data from another

study? If that fails as well, will you then look to apply an arbitrary correlation coefficient of 0.5 and then perform sensitivity analyses? Or will you just use post-intervention only values if baseline values are deemed similar enough? It would be worthwhile stating a little more detail for this.

You bring up important questions. Due to word limit restraints, we referred to the methods in the Cochrane handbook section 6.5.2. in the submitted manuscript. To clarify, we do not expect all studies to report mean differences, rather, this is the summary statistic to be used in the meta-analyses. The mean difference will be calculated from post-intervention measurements or change-from-baseline measurements. Additional data will be sought from authors, initially. Subsequently, obtaining/imputing SDs will be done according to the equations in section 6.5.2.8. I.e. correlation coefficient based from another study if applicable, or by imputing it.

As you mention, we should perform sensitivity analyses for this, which has been added in the sensitivity analyses section.

Pg 12. Line 2-5: *Data extraction (and any needed imputations and calculations) will be done according to recommendations in Cochrane Handbook for Systematic Reviews of Interventions Version 6.2, section 6.5.2.⁴ If needed, additional information from study authors will be sought using e-mail.*

Pg 12, Lines 3-4: You may have instances where there are more than one relevant intervention arm. I would suggest being open to combining groups as per Cochrane guidance. Of course, this should be used sparingly, and if one group is the most relevant, then focus on this group. I suggest adding a sentence in relation to how you will approach these studies.

If there are more than one relevant intervention arm we will include all relevant ones. If applicable we will combine as per Cochrane guidance.

Page 12, line 10-13: *Where multiple trial arms are reported in a single study, we will include only relevant ones. If applicable, we will combine intervention arms or reported subgroups as per the recommendations in Cochrane Handbook for Systematic Reviews of Interventions Version 6.2, section 6.2 and 6.5, respectively.*

Pg 12, heterogeneity: What is the prespecified threshold used for suggesting substantial heterogeneity to warrant subgroup analyses?

We did not pre-specified thresholds for heterogeneity as this can be misleading (Cochrane Handbook, 10.10.2.) and is not as simple as a single value. However, we acknowledge it might also be confusing not to do so, and has therefore added a I^2 value of at least 50%, while this must be interpreted together with P-value from χ^2 test and the confidence interval of I^2 .

Pg 12, line 16-18: *“If we identify substantial heterogeneity ($I^2 \geq 50\%$) we will report it and explore the possible causes by prespecified subgroup analysis. The importance of the I^2 statistic will be interpreted together with the confidence interval and P value from the chi-squared statistic.”*

Pg 12, subgroup analyses: What is the rationale for 20 sessions as a cut-off for intervention duration? I just wondered that with pulmonary rehabilitation being 12-16 sessions long (usually!) that this maybe a more suitable way of guiding the cut-off (i.e., >16 vs ≤16)?

The rationale for 20 session was based on previous ATS/ERS suggestions as a minimum. However, we have considered this and now changed it to 16, which is a more commonly used cut-of today which we also think is reasonable.

I.e. Rochester 2015, <https://iris.unimore.it/bitstream/11380/1073576/2/ATS-ERS%20Policy%20Statement%20on%20PR%20-%20document%202015.pdf>

Pg 12, subgroup analyses: Given your outcomes can be significantly influenced by body mass, would it be possible to include this as a prespecified subgroup? Especially given that sex is also included as a subgroup. Maybe BMI, body mass/weight and/or a measure of fat-free mass could be used?

If there are enough of studies reporting data on this so they can be compared (i.e. normal vs. overweight) we will do this subgroup analysis. We suggest BMI since a measure of fat-free mass might exclude many studies, although it would have been useful.

Pg 12, sensitivity analyses: In relation to my earlier comment about correlation coefficients, it will be important to perform sensitivity analyses to make sure these do not significantly impact the pooled estimates. So, I would recommend stating that sensitivity analyses will be performed for studies where the SD of change has been imputed using a correlation coefficient.

We agree and have added this information. Page 13, line 10-15:

“If we impute standard deviations using correlation coefficients in the synthesis, sensitivity analyses will be performed with different values of the correlation coefficient to determine if the overall results are robust to the imputed correlation coefficients. When appropriate, we will compare the results from an absolute change from baseline versus a percentage change from baseline meta-analysis.”

Reviewer: 3

Dr. Rui Vilarinho, Polytechnic Institute of Porto

Comments to the Author:

Thank you for asking me to review this manuscript presenting a protocol for a systematic review and meta-analysis of physiological responses and adaptations to exercise training in people with or without chronic obstructive pulmonary disease. As strengths of this manuscript, this protocol will be reported according to the Preferred Reporting Items for Systematic Reviews and Meta-Analyses Protocols, is registered at PROSPERO, and the risk of bias assessment is mentioned.

We thank you for your thorough review of this protocol with your valuable time.

General comment: Throughout the manuscript, several terms are used to describe the same information (e.g, "physical capacity", "cardiorespiratory fitness", "physical capacity"). Although there are similarities between them, these terms refer to different definitions. Can authors explain the reasons for using several terms and try to uniformize the information?

We have uniformed the terms to “cardiorespiratory fitness” which is the more correct term in this context.

KEYWORDS

1) The keyword "exercise training" is duplicated (Title and Keywords). Can the authors include new keywords and different from the title to enhance the search in the future publication of the protocol?

Thank you for the suggestion, we have removed “exercise training” and added “endurance training” and “resistance training”.

INTRODUCTION

2) Page 5, line 15. Reference 6 may not be sufficient for the information that is presented. Please consider including more pertinent references such as:

Machado A, Marques A, Burtin C. Extra-pulmonary manifestations of COPD and the role of pulmonary rehabilitation: a symptom-centered approach. *Expert Rev Respir Med.* 2021 doi: 10.1080/17476348.2021.1854737.

This is an excellent reference for this information and we have added it to the manuscript.

3) According to the studies cited in the Introduction section (lines 52-60), it seems that a variety of exercise interventions could be an important barrier/limitation for this systematic review meta-analysis. The authors of this manuscript presented studies that applied "high-intensity training" (reference 31), "multidisciplinary exercise training" (reference 3), and "endurance training" (reference 34). Do authors believe that this information is important to be included in the manuscript as a possible limitation?

The variety of interventions could indirectly become a limitation, leading to too diverse study characteristics, not enabling pooling of studies for meta-analyses. In this case, we will do a narrative systematic review of the available studies. We are aware of this and are discussing it under 'synthesis'. The editor instructed us to remove any discussion not related to the methodology of the paper. In the final publication this will be discussed.

METHODS

4) Throughout the manuscript, "healthy controls" or "people without COPD" are mentioned to present the same population. Can the authors standardize the designation of this population in a single term? However, it is not guaranteed that "healthy controls" and "people without COPD" have the same characteristics. Do authors believe that is important to certify in the studies that will be selected for this systematic review the type and number of comorbidities of "people without COPD" (e.g, no statistical differences in the number of comorbidities or no differences in comorbidity indexes between groups)? Can this information influence the objectives and the further results of the meta-analysis?

We have standardised the term to people without COPD and agree that this is a difficult issue. While healthy controls are often used in the literature, we have chosen not to use this term since they will not necessarily be perfectly "healthy". Even the same term, i.e. "healthy controls" will have a variety of characteristics in the literature. We have added that healthy controls are often used as the term:

Pg 5 line 35-37: *"Types of participants*

Any study including adults ≥18 years of age with a diagnosis of COPD and a group of individuals without COPD (often referred to as "healthy controls") will be considered."

We will not set specific criteria (i.e. number of comorbidities) and in general not exclude studies due to different characteristics of the non-COPD group. However, as you emphasise, this can influence the efficacy of an intervention/response to exercise, and must therefore be considered and discussed in the review. We have added number of comorbidities as a study (participant) characteristics to extract and summarise, which was a good idea.

5) Please clarify how the diagnosis of COPD based solely on spirometry would not misclassify chronic persistent asthma as COPD.

We have changed our wording on this to: "with a diagnosis of COPD" to be more inclusive. A more detailed description could exclude too many studies.

6) How the outcome measures will be selected according to the interventions. For example, it will be used the same outcome measures for neuromuscular electrical stimulation (NMES) and inspiratory muscle training (IMT)?

While some outcomes are irrelevant for some interventions, we do not expect this to be a major issue in this systematic review: Studies will (in general) only measure relevant outcomes. Thus, we have

not linked outcomes to specific interventions. Only relevant outcomes for the specific intervention will be included. For example, quadricep strength could be an outcome for NMES but not for IMT.

7) Page 7 line 13-14. Can the authors include a reference to support this information?

If you are referring to the intervention duration of at least 12 sessions, this was chosen since it is the minimum recommendation in some guidelines. However, on a second thought, we will not have a minimum number of sessions for inclusion (to be a bit more inclusive), only the three weeks duration. This is a common approach that has been proven effective, e.g. in Germany. The more inclusive approach might enable a better subgroup analysis on extensiveness.

8) In outcomes measures: can respiratory exchange ratio (RER) be included?

The RER can indeed be included and has been added.

Furthermore, do authors consider that it is important to select the outcomes measures according to the level of evidence on their measurement properties (according to COSMIN and GRADE), namely for the tests mentioned in "exercise capacity", "functional performance", "cognitive function"?

This is an interesting thought. However, we think that selecting outcomes according to their measurement properties, i.e. COSMIN is outside the scope of this study.

9) Page 7, in Types of outcome measures subsection, it is important to clarify the definitions of exercise capacity and maximal exercise performance. They are not the same? Can the 6MWT and step tests be considered more appropriate as physical tests to assess functional capacity?

The tests in the "exercise capacity" are not maximal efforts in the same manner as the maximal exercise performance test (as a ramp/incremental test to exhaustion). However, you are correct that the 6MWT is better defined as functional capacity which we clarified in the section. We have changed our terminology a bit here. Exercise tolerance is used instead of exercise capacity, and we made "Functional performance" as a separate bullet:

As seen on page 6: *Functional performance: e.g. 6-minute-walk-test, 5-time sit-to-stand test, 30-sec sit-to-stand and 1-minute sit-to-stand test.*

After some consideration, we still have the ESWT and step tests under "exercise tolerance", which we think is reasonable and commonly used.

10) Page 13. Subgroup analysis and investigation of heterogeneity subsection: reasons for "duration of intervention \geq 20 sessions compared to $<$ 20 sessions"?

The rationale for 20 session was based on previous ATS/ERS suggestions as a minimum. However, we have reconsidered this and now changed it to 16, which is a more commonly used cut-of today which we also think is reasonable.

REFERENCES

11) Errors were found in some references. For example, reference 34.

Thank you for noticing this. This error, and a few other small errors (titles, journal abbreviations), have been corrected.